# Does electronic consent improve the logistics and uptake of HPV vaccination in adolescent girls? A mixed-methods theory informed evaluation of a pilot intervention

Tracey Chantler [ORCID],[1] Ellen Pringle,[2] Sadie Bell,[1] Rosie Cooper,[3] Emily Edmundson,[4] Heidi Nielsen,[4] Sheila Roberts,[4] Michael Edelstein [ORCID],[2] Sandra Mounier-Jack[1]

TC and EP are joint first authors.

For numbered affiliations see end of article.

**Correspondence to**
Dr Tracey Chantler;
tracey.chantler@lshtm.ac.uk

## ABSTRACT

**Objectives** To evaluate the usability and acceptability of an electronic consent pilot intervention for school-based immunisations and assess its impact on consent form returns and human papilloma virus (HPV) vaccine uptake.

**Design** Mixed-methods theory-informed study applying qualitative methods to examine the usability and acceptability of the intervention and quantitative methods to assess its impact.

**Setting and participants** The intervention was piloted in 14 secondary schools in seven London boroughs in 2018. Intervention schools were matched with schools using paper consent based on the proportion of students with English as a second language and students receiving free school meals. Participants included nurses, data managers, school-link staff, parents and adolescents.

**Interventions** An electronic consent portal where parents could record whether they agreed to or declined vaccination, and nurses could access data to help them manage the immunisation programme.

**Primary and secondary outcome measures** Comparison of consent form return rates and HPV vaccine uptake between intervention and matched schools.

**Results** HPV vaccination uptake did not differ between intervention and matched schools, but timely consent form return was significantly lower in intervention schools (73.3% vs 91.6%, p=0.008). The transition to using electronic consent was not straightforward, while schools and staff understood the potential benefits, they found it difficult to adapt to new ways of working which removed some level of control from schools. Reasons for lower consent form return in e-consent schools included difficulties encountered by some parents in accessing and using the intervention. Adolescents highlighted the potential for electronic consent to by-pass their information needs.

**Conclusions** The pilot intervention did not improve consent form return or vaccine uptake due to challenges encountered in transitioning to new working practice. New technologies require embedding before they become incorporated in everyday practice. A re-evaluation once

## Strengths and limitations of this study

- ► The use of a theory-informed mixed-methods study design allowed us to measure the effect of a pilot e-consent intervention on immunisation performance and identify mechanisms that facilitated or impeded implementation.
- ► The study design allowed us to account for schools, nurses, data managers, parents and adolescents' experiences of using the e-consent technology in this evaluation.
- ► Data limitations include the lack of interviews with school staff to complement the feedback forms and not being able to access and interview parents, who found the intervention more challenging to use.
- ► This evaluation was limited to one cycle of implementation, which did not allow us to account for the time it takes for new practices to become embedded. The number of schools who declined to pilot the intervention was also not recorded.

stakeholders are accustomed with electronic consent may be required to understand its impact.

## INTRODUCTION

Vaccination against human papilloma virus (HPV) provides long-term protection against cervical and other cancers and genital warts.[1 2] In England, HPV vaccination (two doses 6 months apart) is offered to adolescents aged 12–14 years in the school-based immunisation programme, which is delivered by nurse-led immunisation teams. While the school delivery model is widely accepted,[3] it raises logistical challenges regarding obtaining parental consent. Immunisation teams and schools identify all students, who are eligible for vaccination and then students are typically given a paper consent form that their parents/legal guardians need to sign

BMJ

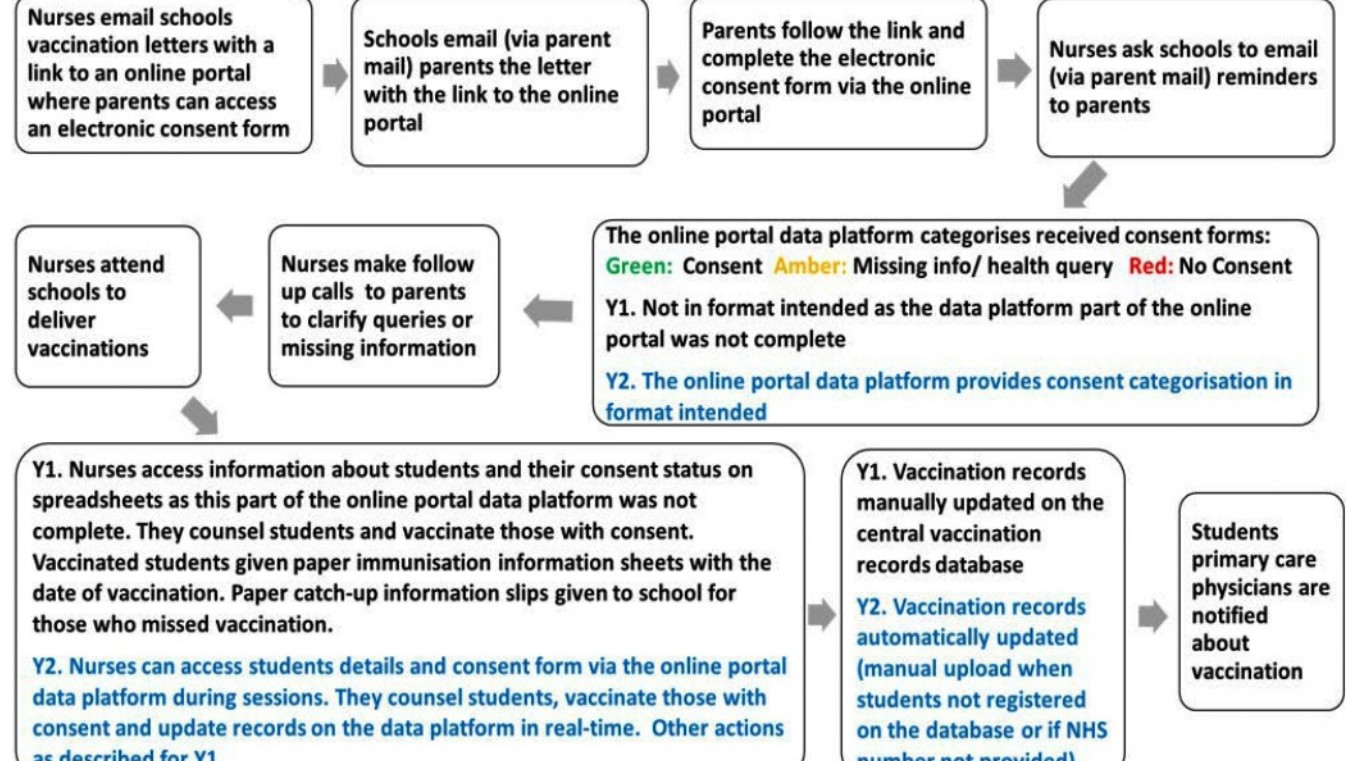

**Figure 1** The electronic consent intervention.

indicating their acceptance or refusal of vaccination. The passage of the form from teachers to students to parents and back again can impact on form return and have a detrimental effect on vaccine uptake.[4–7] Follow-up by school staff and immunisation teams improves uptake but is resource intensive.[8 9] In the context of the drive towards a 'paperless National Health Service (NHS)',[10] there is increasing interest in the potential for technological solutions, such as electronic consent (e-consent). We evaluated the usability and acceptability of a pilot e-consent intervention from the perspective of parents and adolescents, health professionals and schools, and assessed its impact on consent form returns and HPV vaccine uptake.

### The electronic consent intervention

The e-consent intervention was developed by Hounslow and Richmond Community Healthcare NHS Trust (HRCH) with the support of a software development company in 2017/2018 and piloted in their adolescent girls' vaccination programme in June/July 2018. At the time of the study, HRCH was responsible for administering this vaccination programme in secondary schools across eight boroughs in South London.

The e-consent intervention consisted of an online portal with an e-consent form and a data platform and related implementation procedures (figure 1). Functionally it aimed to: (1) give parents access to an online portal with information about the vaccination programme where they could register their adolescent and agree to or decline HPV vaccination; (2) give nurses electronic

access to the portal to facilitate screening and enable them to update records during immunisation sessions; (3) enable automatic updating of central vaccination record databases. Parts of the online portal and data platform (specifically those relating to points 2 and 3) were not fully functioning before the intervention was first used in June 2018. Hence modifications had to be made to the way nurses screened students' information and consent forms before and during immunisation sessions. Figure 1 differentiates what happened in Year 1: June/July 2018 and in Year 2: June/July 2019 (blue text).

In the quantitative part of evaluation, we focused on year 1; the measurement of outcomes and impacts. In the qualitative part of the evaluation we captured how year 1 experience informed adaptations to the intervention prior to reuse in year 2.

### METHODS
### Study design

This was a mixed-methods theory-informed evaluation study which used a 'Theory of Change' (figure 2) as an evaluation framework. Quantitative methods were used to assess whether the pilot intervention increased consent form return and the uptake of the first dose of HPV vaccine in adolescent girls in June/July 2018 (year 1). The steps involved in implementing the e-consent system, people's experiences of these, the interactions between inputs, activities, pathways and outputs, outcomes and

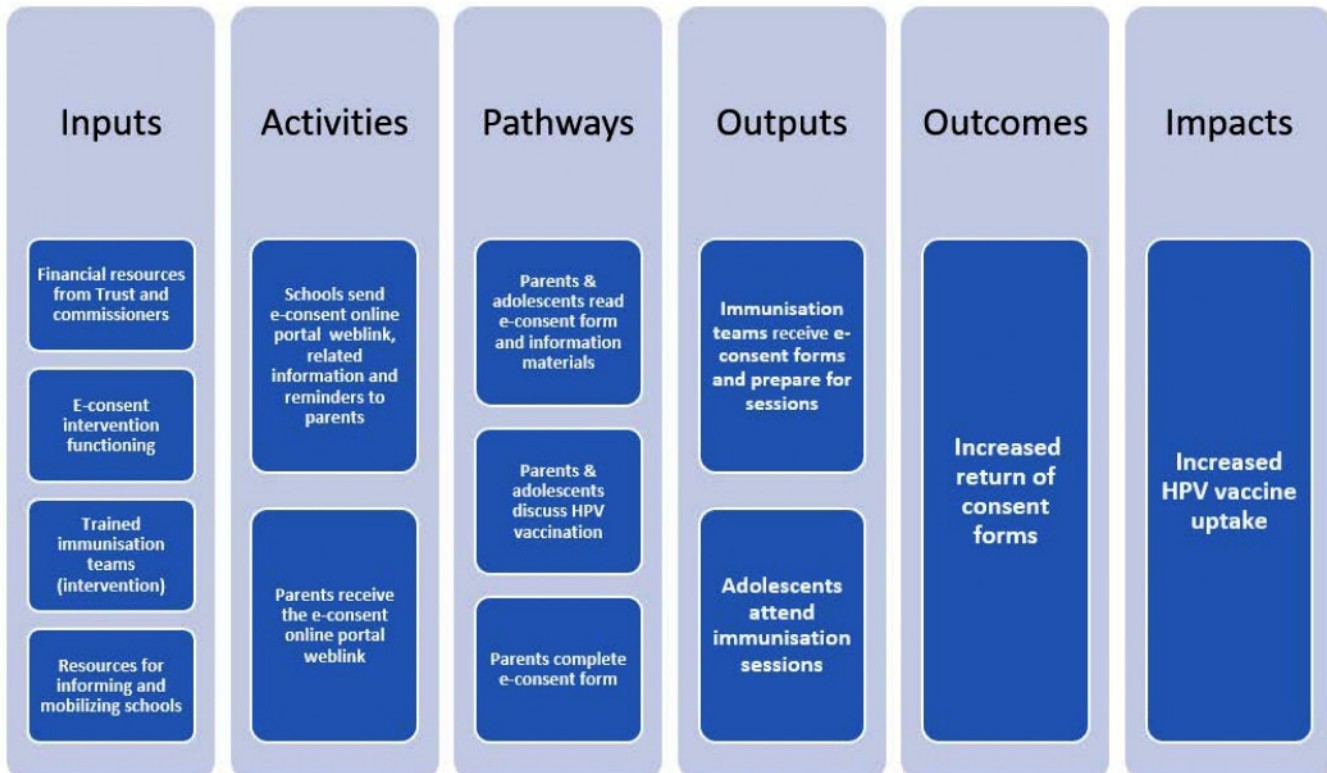

**Figure 2** Theory of change for the e-consent intervention.HPV, human papilloma virus.

impacts were investigated using qualitative methods in year 1 and Year 2. The reason that the quantitative analysis only focused on year 1 was that the schools that received the intervention between year 1 and 2 changed.

### School selection

The e-consent intervention was piloted in 14 secondary schools in seven South London boroughs in June/July 2018 (year 1). Schools selection was purposive with the aim of including schools that differed in terms of denomination (private, state, grammar), type (mixed, single sex), sociodemographic, size, vaccination uptake and level of support to the programme.

Schools were divided into low, medium or high based on the proportion of pupils receiving free school meals and with English as an additional language. Each e-consent school was matched, as closely as possible, to a paper consent school in the same tertiles for both characteristics.

### Quantitative methods and analysis
#### Data collection (year 1)

At each vaccination session nurses completed a 'tally sheet', with details of the consents received prior to or during the session, any absences and the number of vaccinations given. These tally sheets were completed for paper and e-consent schools.

In addition, we extracted the name of the school, date and time of consent form completion and type of consent (agreement by parent or adolescent self-consent or decline) for each consent form in the e-consent system. This non-identifiable information was combined with tally sheet data, which was manually transcribed into MS Excel. Where there were discrepancies or missing data this was checked with the immunisation teams and in the case of the e-consent schools the system data was used in preference to the tally sheet data.

#### Data analysis

Characteristics of the paper and e-consent schools were described in terms of proportion of pupils receiving free school meals, speaking English as an additional language, ethnicity and characteristics of the schools (religious affiliation and state/privately funded), using data from the Office of National Statistics. (REF: https://www.gov.uk/government/statistics/schools-pupils-and-their-characteristics-january-2018).

For both paper and e-consent schools, we calculated: (1) the proportion of the pupils for whom a consent form had not been returned prior to the vaccination session, (2) the proportion of pupils vaccinated at the planned session and (3) the proportion of pupils for who a 'yes' consent was received (prior to or on the day of the planned vaccination session).

We compared e-consent schools with paper consent schools in terms of timely form return, outcome of consent and vaccination uptake using Kruskal-Wallis tests for statistical significance. Where data were missing from a school that school and its matched pair were excluded from the analysis. All analyses were done in MS Excel and Stata V.15.

## Qualitative methods & analysis
### Data collection
Data collection occurred in June–December 2018 (year 1) and June–July 2019 (year 2). The first period coincided with and followed the first year of e-consent implementation, the second occurred during the use of e-consent in a different subset of schools.

### June–July 2018 (year 1)
*Observations of e-consent school HPV immunisation sessions*
Members of the evaluation team (RC, TC and SM-J) accompanied immunisation teams during seven e-consent school HPV immunisation sessions to obtain a contextual understanding of the implementation of the intervention. The evaluators documented what worked well and what if any problem solving was applied during sessions.

*School feedback forms from paper and e-consent schools*
School staff involved in organising immunisation sessions were asked to complete a feedback form after the HPV immunisation sessions (online supplemental additional file 1). Forms included questions about the organisation of immunisation sessions and the usability and acceptability of e-consent and paper consent. Seven e-consent and seven paper consent schools completed forms.

*Individual and peer group interviews with HRCH staff*
These semistructured interviews (SSI's) involved HRCH staff who were responsible for implementing the intervention. The programme manager was interviewed individually, the two data specialists together and member of four immunisation teams in peer groups of two to four participants (12 interviewees) at their respective offices. In total 15 members of staff took part in hour long interviews.

*Interviews with parents and adolescents from e-consent schools*
These SSIs involved parents and adolescents who had used the e-consent system. Parents/legal guardians were asked to indicate their willingness to be contacted by researchers in the e-consent form (ticked a statement). Parents who responded positively were invited by email to take part in an interview to discuss their experience of using the e-consent system. A total of 12 interviews were conducted, nine with HPV vaccine acceptors and three with decliners. Four interviews were conducted in family homes, seven by telephone, and one by Skype video. Adolescents participated in five interviews.

### June–July 2019 (year 2)
Results from data collected in year 1 informed the implementation of the e-consent intervention in 22 schools located in South London in the 2018/2019 HPV vaccine programme. Only one of these schools had taken part in the 2018 pilot, the reasons for using the improved intervention in this school and not the other pilot schools was mainly due to the scheduling of the HPV vaccination sessions. The Trust also wanted to expose additional schools to the e-consent system. To reflect the changes made to the intervention and its implementation two group interviews with HRCH staff (one with nine and the other with five members of staff) and one focus group discussion (FGD) with eight adolescent girls (from a school that used the e-consent intervention in 2019 only) were conducted.

The topics covered in the staff, parent and adolescent SSIs and adolescent FGD are summarised in table 1 (see online supplemental additional file 2 for topic guides).

### Data analysis
The SSIs and FGD were audio recorded with participant's permission and transcribed anonymously. Transcripts, observational field notes and school feedback forms were downloaded into a qualitative data analysis management software program (NVivo V.12). We adopted a thematic analytical approach which combined semideductive mapping of data to the 'input', 'activities', 'pathways', 'outputs', 'outcomes' and 'impacts' depicted in the Theory of Change (ToC) and inductive open coding to capture emerging themes.[11] We sought to account for the interdependence of ToC categories and real-life experience of managing organisational change, which do not always progress from inputs—impacts in a seamless linear manner.

### Patient and public involvement
This evaluation was designed with members of HRCH who were responsible for coordinating the implementation of the e-consent intervention. This collaboration supported the iterative and reflexive development

**Table 1** Topics covered in interview

| Staff interviews | Parent and adolescent interviews |
|---|---|
| ► Experience of managing the administration of the school-based vaccination programme. | ► Views on adolescent vaccination. |
| ► Experience of obtaining consent in school-based vaccination programme. | ► Understanding of adolescent vaccination. |
| ► Acceptability and usability of the e-consent system. | ► Experience of providing consent for the HPV vaccine. |
| ► Interaction between schools and immunisation teams. | ► Acceptability and usability of the e-consent system. |
| ► Adolescent self-consent. | ► Teenagers experience of immunisation in school. |
| ► Reflections on use of the e-consent system. | ► Vaccine programme communication. |
| | ► Views on adolescent self-consent. |

HPV, human papilloma virus.

**Table 2** Characteristics of paper and e-consent schools† ‡

|  | Paper consent schools (n=13) | E-consent schools (n=13) | P value* |
|---|---|---|---|
| % (range) of children eligible for free school meals | 17.3 (1.5–52.1) | 17.3 (1.5–43.2) | 0.84 |
| % (range) of children with English as additional language | 30.9 (5.5–53.6) | 32.4 (9.1–59.2) | 0.72 |
| % (range) of children of white British ethnicity | 30.9 (1.5–69.6) | 34.6 (6.3–65.7) | 0.63 |

*Refers to comparison of each characteristic using Kruskal-Wallis test.
†Characteristics information was not available for the two private schools.
‡Data from Schools, pupils and their characteristics: January 2018, Department of Education.

of the intervention, which is essential for longer-term integration.

## RESULTS
### Quantitative
#### Participants
Twenty-eight schools (14 paper and 14 e-consent schools) comprising 3219 girls (1733 in paper consent and 1486 in e-consent schools) were included in the study. Of those schools, 26 were state and 2 were private schools. Twenty-one of the schools had no religious affiliation, three were Roman Catholic (all paper consent), three Church of England (two e-consent, one paper consent) and one was another Christian faith school (paper consent).

The proportion of pupils eligible for free school meals, with English as an additional language and students' ethnicity profile was similar between the e-consent and paper consent schools (table 2).

#### Return of consent forms ahead of session
Overall, 83% of consent forms (paper or e-consent) were returned prior to the vaccination session. However, among the 22 matched schools where this data was available, compared with paper schools timely (prior to the planned session) return was lower in the e-consent schools (73.3% vs 91.6%, p=0.008).

#### Outcome of consent
There was no statistically significant difference in the proportion of pupils for whom a 'yes' consent was received (prior to or on the day of the session) between the paper (n=14) and e-consent (n=14) schools (85% in e-consent schools, 83% in paper consent schools, p=0.89).

#### Vaccination uptake
There was no statistically significant difference in the proportion of pupils that were vaccinated at the scheduled vaccination session between the paper (n=14) and e-consent (n=14) schools (80.6% vs 81.3%, p=0.93). These figures did not include those who were absent

on the day and vaccinated later (Paper consent would have been used in the 14 control schools at school and community based catch up clinics. All parents in e-consent schools would have received the weblink to the e-consent portal and in most cases will have used this link to complete the consent form. Some students who attended e-consent school catch up clinics did not have a signed e-consent form. In those instances, the nurses would phone parents during clinics or obtain adolescent self-consent if appropriate. In the case of community-based catch up clinics parents/guardians and students would usually attend together, if the e-consent had not been completed nurses would either complete the e-consent form at the clinic with the parent and student, if Wifi/technology was available or complete a paper consent form). The final vaccine uptake across all the schools was over 86%.

### Qualitative
The results of the ToC thematic analytical mapping are presented under relevant headings ('input', 'activities', 'pathways', 'outputs', 'outcomes'), followed by an overarching theme on managing change.

### Inputs
#### Resources and training
There was a 'buzz' about the development of the e-consent intervention, its potential to streamline consent and facilitate safer data collection and NHS England had provided funding for HRCH to pilot it during the 2017/2018 HPV vaccine programme. Immunisation team members were positive but expressed reservations about not able to review paper consent forms prior to immunisation. Due to tight deadlines only one orientation session took place before the e-consent intervention was introduced, which meant that the bulk of learning happened on the job.

> I think as well, it was probably four days before our first session, we didn't know what we were doing… so I do feel we are running before we can walk. (Immunisation Team 2)

#### Intervention not fully operational
The data platform component of the online portal was not operational prior to implementation. Parents were able to access and complete the e-consent form, but the immunisation teams could not review student's online consent forms or upload data during immunisation sessions. Instead large (A3) paper sheets with information about who had provided consent were prepared by the data managers. The sheets were difficult for to decipher during busy sessions and nurses were less able to prepare cohort figures and tally sheets in advance.

> It was an anti-climax not being able to use the laptops and still have a paper sheet in front of me. (Immunisation Team 4)

## Mobilising and resourcing schools

There was limited time to collaborate with schools before the vaccination programme, although all e-consent schools were guided on how to disseminate the weblink. A few schools in two inner city boroughs declined to use the intervention due to concerns about: (1) pre-existing barriers to electronic communication with parents, (2) whether a change to consent processes would reduce the return of consent forms and uptake of HPV vaccine and (3) adapting to a new way of working.

School immunisation link staff reported a 'loss of control', for example, they could no longer see 'who had said yes, and who had said no', which restricted their ability to follow-up unreturned forms. Paper-consent schools could monitor this directly by counting forms, but with e-consent schools immunisation teams had to check parental responses and tell schools which families had not replied.

## Activities
### Dissemination of the online e-consent portal weblink to parents

E-consent schools used different means (parent mail, email, school website, newsletters, letters) to send parents the portal weblink. Blanket reminders were mainly sent electronically, unless immunisation teams provided schools with details of non-responders. In this case follow-up could be more targeted and involve text messages and phone calls as wells as emails. One school used a translator to engage parents who did not understand the consent process due to language barriers. Another school was not willing to send out emails and asked the immunisation team to provide them with printed letters referring to the weblink to send to parents.

Of the seven e-consent schools who completed the feedback form, four were positive about the intervention and how it had been implemented stating that it had reduced their workload. Another school was mainly positive but noted that some parents had found the e-consent form difficult to access, another reported that their parent cohort had found the system very difficult to access and use, and the last school was the one who had used letters to disseminate the weblink.

## Pathways
### Navigating the e-consent form and related information

The e-consent form included links to an HPV vaccination leaflet. However, none of the interviewees had downloaded or read this leaflet for the following reasons: accessed information elsewhere, already sufficiently informed, older daughter vaccinated, positive about vaccination.

Proactive information seeking involving a wider range of sources was more common in families who were vaccine hesitant. Their concerns, which in some cases resulted in vaccine refusal, included the following: (1) compatibility of vaccination with pre-existing illness, (2) unknown side effects (eg, hormonal interferences while daughter establishing her cycle), (3) targeting of adolescents due

to their ethnicity (nurses explained that some parents of Black-Caribbean origin were sceptical of any state intervention, like vaccination) and (4) necessity of vaccination for adolescents who are not sexually active. Parents who were more confident about vaccines restricted their information seeking to NHS sources and suggested that a 'road map' to adolescent vaccination could be useful.

Adolescents reported a variety of information seeking behaviours. Some just accepted HPV vaccination as 'something that needs to be done' and felt reassured that it was recommended by the NHS: 'I think because it's like by the NHS—it kind of gives it validation.' (Adolescent 9—Yes). Others wanted the HPV vaccine leaflet to include more information about HPV and related health risks and vaccine side effects, so that they did not panic if they experienced any of these.

In the FGD students expressed a preference for paper leaflets and discussed how the e-consent could bypass them: '…because like if it's emailed, like your mum doesn't have to share it with you. And like if I have something done like an injection, I'd like to know what's going on and when. But like she filled out the form without like telling me, so like if they'd been given out in school then I could have read it and see what's happening.'

These FGD participants wanted to have the opportunity to discuss vaccination with their parents; they were not seeking autonomy in decision making rather wanted a degree of joint responsibility.

> I wouldn't like to be given the option to like not to have the injection done…so I'm kind of glad that my mum just decided like on her own. But I would have liked her to talk it through with me… (FGD participant)

Parents approaches to talking to their daughters about having an HPV vaccination are summarised in box 1.

### Using the e-consent intervention

The parents we interviewed found the system easy to use and usually completed the form as soon as they received

---

**Box 1   Parents approaches to discussing human papilloma virus vaccination with adolescent daughters**

Very limited discussion
► Did not talk to daughter in detail since vaccination is a parental decision.
► Talking too much about pending vaccination may induce anxiety, especially if needle phobic.

Heads up as what to expect
► Ensured daughter knew what to expect; this exchange usually occurred shortly before the immunisation session.
► Offered to answer any questions depending on adolescents' desire to know more.

More in-depth discussions
► Joint decision making between parents and adolescents about the important of vaccination, this sometimes involved accessing additional information.

it. A few parents would have liked an email confirmation after they had completed the e-consent form.

> I thought it was very easy. I think you're probably going to get more responses that way from parents in this day and age. However, the downside is obviously you may not get that chance to discuss it. (Parent 5—Yes)

According to feedback from nurses and schools not all parents found the intervention easy to access or use. Language barriers accounted for some difficulties, but practical issues also played a role, for example, some parents had not signed up for the school parent mail system hence did not receive the weblink. Nurses also received a significant number of calls from parents reporting that the weblink would not open or that webpages froze. During an immunisation session one student stated: 'my dad said I should have the vaccine, but he did not understand the whole google business about it'. In some instances, the weblink closed a few days prior to an immunisation session to give immunisation teams time to screen student information prior to sessions. This resulted in some parents who had missed the last sign-up date sending in written notes to confirm their consent/non-consent for vaccination.

## Outputs
### Nurses access to e-consent forms and student information
The immunisation sessions at e-consent schools were affected by nurses not being able to access electronic information about students before and during sessions. The lower return of consent forms in e-consent schools also resulted in nurses reporting that they had to contact more parents than usual during immunisation sessions to obtain verbal consent. This had implications for the nurses' workload distribution and the length of sessions. To manage these challenges immunisation teams increased the number of nurses and administrative assistants who attended e-consent school immunisation sessions.

> …we had 80 consent forms outstanding at a big school. But, normally, if you only have a couple it's fine. It meant us was making calls all morning, it took a nurse out of immunising to be able to do that, so that did have a big impact. (Immunisation Team 2)

Conducting phone calls during sessions was not straightforward. First, nurses had to rely on students (if they had phones with them) or staff to help them access correct contact details. Second, immunisations sessions were busy and noisy which impeded communication and privacy. Thirdly, it was not always possible to reach parents who were at work or out of the house during day-time hours. If parents were uncontactable the nurses assessed if students who wanted to be vaccinated had sufficient maturity and intelligence to understand and appraise the nature and implications of the proposed vaccination.[12 13] This process was time-consuming and not all nurses felt comfortable about vaccinating without verbal or written parent permission.

### Transition: adapting to change and iterative development
The initial 'buzz' about the e-consent intervention decreased over time. While some staff remained positive and receptive to the implementation coordinators enthusiasm and vision, others expressed a sense of half-heartedness about having to adapt quickly from a known way of obtaining consent, although with flaws (eg, cost of paper, mileage clocked up in collecting paper consent forms from schools), to a new technology enabled way with some functional limitations in year 1 (see figure 1).

In the HRCH staff group interviews conducted in July 2019 members of the immunisation teams reflected on lessons learnt from their experience of transitioning to a 'brand new way of working' over the past year. Key learning points from an internal organisational perspective were: (1) adopt right pace of progress when introducing new interventions with have several components, (2) be clear about which part of a multi-component intervention is being piloted and implemented (eg, in year 1 it was primarily about the e-consent form), (3) importance of timely communication, quick thinking, and flexibility when things do not quite go to plan.

In terms of school engagement HRCH staff emphasised the importance of close collaboration to devise appropriate means of consent in different educational and social contexts. The right balance of responsibility between schools and immunisation teams needs to be negotiated to maintain positive working relationships and ensure that adolescents can access essential vaccines.

> I would also say the idea of just changing to e-consent… schools need different things… it is really important to work with the school and a make sure that they are happy with everything and it suits that school, because some schools it might just not suit right now. It might suit them in a couple of years, but right now it just doesn't work. (HRCH immunisers group interview, July 2019)

HRCH made several changes to the implementation of the e-consent intervention in year 2 (2018/2019) informed by evaluation findings (box 2).

---

**Box 2  E-consent intervention implementation changes between Yr1 and Yr2**

► Taking more time to engage (emails, phone calls and meetings) with schools in preparations to find the right level of involvement.
► Ensuring students receive a paper copy of the human papilloma virus adolescent programme leaflet produced by Public Health England in addition to information provided in the e-consent form.
► Pushing more for assemblies and contact with adolescent girls prior to the immunisation sessions.
► Providing ongoing training and mentoring of immunisation teams on use of the intervention.

---

## DISCUSSION

The pilot e-consent intervention had no detected impact on vaccine uptake and did not improve the return rate of consent forms in its first year of implementation. This could be interpreted as a negative result, however this would fail to account for the embedding process required before new technologies become (or do not become) routinely incorporated into everyday practice,[14] and inherent difficulties in achieving high vaccine uptake in urban centres. Our findings demonstrate that the transition from paper to e-consent was not straightforward, and that staff and schools, though mainly open to change, took time to adapt. The transition was hampered by the reduced functionality of the information technology capability (intervention fidelity), limited staff training and engagement of schools. It is likely that this e-consent intervention would need to be used for more than 1 year to achieve any benefits to process efficiency and vaccine uptake.

This study starts to address an evidence gap and demonstrates how real-time evaluation can support the iterative development of interventions essential for their longer-term integration. Key limitations include the lack of interviews with school staff to complement the feedback forms and not being able to access and interview parents who found the intervention more challenging to use.

The 'normalisation process theory' argues that new practices become routinely embedded as the result of people working, individually and collectively, to enact them. Enactment is promoted or inhibited through the operation of four mechanisms; coherence (sense making), cognitive participation (individual buy-in), collective action (joint effort) and reflexive monitoring (user appraisal).[15] Sense making requires users to have a shared understanding concerning the purpose of an intervention, their responsibilities in its implementation, its potential benefits and the differences to existing practice.[16] In our study, immunisation teams recognised the potential value of the intervention but were not clear about their specific responsibilities and schools had to make sense of the change in their role in consent logistics. User buy-in is defined as user's agreement to try out a new way of completing a task and their willingness to drive and sustain the implementation of a new intervention.[15] HRCH coordinators encouragement, ability to adapt when problems arose was critical to sustaining momentum throughout the implementation phase. Most schools bought into the concept of technology-enabled consent yet their willingness to sustain this change could have benefited from earlier collaboration between immunisation teams, head teachers and link staff. Parents and student's engagement with the e-consent intervention was variable. Parent interviewees accessed the online portal and were able to complete and submit the e-consent form with ease. However, the significant difference in consent form return rate between e-consent and paper consent schools (73.3% (n=11) vs 91.6% (n=11), p=0.008) response rate (73.3%) indicated that this was not the case

for all parents. Of additional concern was that the e-consent intervention inadvertently bypassed some adolescents' information needs and related opportunity to talk to their parents about HPV vaccination. Collective action was evidenced when immunisation teams and schools worked together to make the follow-up of e-consent form non-response more targeted. Reflexive monitoring was facilitated by this real-time evaluation although more coproduction activities with HRCH staff, parents, school link staff and adolescents need to be integrated in subsequent evaluations.[17 18]

There is a need to streamline consent processes for adolescent immunisation and ensure that parents and adolescents are fully informed about these preventative measures. Electronic consent could be an effective option to achieve this. However, it needs to be tailored to specific contexts, and parents, schools and adolescents need to provide input in the development, implementation and evaluation of such technological interventions. Evaluations also need to factor in the time needed for new working practices to be fully integrated. Our experience suggests that to gain a complete and accurate assessment of the impact of new interventions evaluations need to collect data over more than one cycle of implementation. Introducing change that affects different actors requires all stakeholders to understand, buy in and work together in refining and coproducing complex behavioural interventions.

**Author affiliations**
[1]Department of Global Health and Development, Faculty of Public Health & Policy, London School of Hygiene and Tropical Medicine, London, UK
[2]Department of Immunisation, Hepatitis and Blood Safety, Public Health England, London, UK
[3]University Hospitals of Derby and Burton NHS Foundation Trust, Derby, UK
[4]Hounslow and Richmond Community Healthcare NHS Trust, Teddington, UK

**Acknowledgements** We are grateful for the information shared freely by interview participants, schools and the Hounslow and Richmond Community Healthcare NHS Trust.

**Contributors** TC and SM-J designed the study with input from SR, EE, HN, ME and RC. EP led the quantitative data analysis with support from RC and SB. TC collected and led the analysis of the qualitative data with support from SB. All authors discussed the preliminary findings and contributed to and critically reviewed the manuscript.

**Funding** The research was funded by the National Institute for Health Research Health Protection Research Unit (NIHR HPRU) in Immunisation (Grant reference: HPRU-2012–10096) at the London School of Hygiene and Tropical Medicine in partnership with Public Health England (PHE).

**Competing interests** TC, SB and SM-J report that they were in receipt of funding from the National Institute of Health Research while conducting this research. ME, RC and EP worked for Public Health England for the duration of this research.

**Patient consent for publication** Not required.

**Ethics approval** The research published in this manuscript was approved by the PHE Research Support & Governance Office (Ref: NR0131) and the London School of Hygiene and Tropical Medicine Observational/Interventions Research Ethics Committee (Ref: 15839).

**Provenance and peer review** Not commissioned; externally peer reviewed.

**Data availability statement** Data are available on reasonable request. Data collection tools are provided as online supplemental documents and some anonymised data are available on request via the LSHTM public repository.

**ORCID iDs**
Tracey Chantler http://orcid.org/0000-0001-7776-7339
Michael Edelstein http://orcid.org/0000-0002-7323-0806

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
