## [Reviewer comments · BMJ Open]

ARTICLE DETAILS

TITLE (PROVISIONAL)	DOES ELECTRONIC CONSENT IMPROVE THE LOGISTICS AND UPTAKE OF HPV VACCINATION IN ADOLESCENT GIRLS? A MIXED METHODS THEORY INFORMED EVALUATION OF A PILOT INTERVENTION
AUTHORS	Chantler, Tracey; Pringle, Ellen; Bell, Sadie; Cooper, Rosie; Edmundson, Emily; Nielsen, Heidi; Roberts, Sheila; Edelstein, Michael; Mounier-Jack, Sandra

VERSION 1 – REVIEW

REVIEWER	Suzanne Audrey University of Bristol
REVIEW RETURNED	30-Apr-2020

GENERAL COMMENTS	This paper reports on an important topic, and gives insights into electronic consent for the HPV vaccination programme. The paper is generally well written (I have noted some minor typing errors below) and makes a contribution to knowledge. However, I do have some concerns which I believe should be addressed before publication. - it was not clear to me why the statistical analyses are limited to one cycle, and did not give results for the second year. This is acknowledged as a weakness by the authors and I suspect there is a valid explanation but it was not clear to me on reading the manuscript.- the authors indicate e-consent was 'piloted' in June/July 2018 and the quantitative results relate to that year. This almost suggests that we are presented with the results of a pilot study, rather than an evaluation.- related to the points above, were data on consent returns and uptake collected for June-July 2019? If so, could these be presented? If not, it would be helpful to be clear why these data were not collected and analysed, as the authors indicate there were a number of problems with the e-consent which may have been less marked as the new system became more familiar and some of the teething problems were addressed.- the authors indicate a 'sub-set' of schools took part in June-July 2019. More information on why only a sub-set were involved would be helpful- the authors indicate they could only measure the difference in the return of consent forms for 22 matched schools. It would be helpful to explain why.- in relation to vaccine uptake, the authors indicate that those who were absent on the day were vaccinated later bringing the final update rate up from around 80% to 86%. Do we know if the 6% used e-consent or paper consent?
--

	 - the authors indicate 'a few schools declined to use the intervention due to concerns about communication with parents'. How many schools? Could the reasons be expanded upon as it is an important issue in relation to e-consent. - the authors indicate that, as a result of some of the lessons learned, changes were made to the e-consent intervention, but we do not know if these changes meant e-consent was more effective. This comes back to the issue of whether this study is better described as a pilot study rather than an evaluation. - the authors indicate the study addresses an evidence gap. It certainly raises some key issues but I'm not sure if the evidence gap is fully addressed. I spotted typing errors in the abstract and on pp. 6,13, 17. The article would benefit from proof reading to address these minor issues.
--	--

REVIEWER	Cecilia Trucchi A.Li.Sa., Liguria region, Italy
REVIEW RETURNED	04-May-2020

GENERAL COMMENTS	In addition to the abovementioned comments, I suggest the Authors to address the following minor issues:  - the electronic consent intervention was addressed to adolescent girls and to their parents: what about the gender neutral HPV immunization strategy in UK? - the matching between healthcare and schools data to verify that all subjects for whom the HPV vaccine is recommended were invited to receive HPV vaccine could be better clarified and deepened - the "outputs" section introduces the concept of "maturity and intelligence" assessment of students whose parents were uncontactable: discussing the role of an adolescent "informed assent" in addition/alternatively to the informed consent of parents/tutor could be of interest, since many countries includes these procedures in their immunization process; furthermore, the informed consent's collection of one or both parents/tutors should be better clarified, finally, what about the managing of refusal? Were parents invited to sign an "informed dissent" or similar? - the Results section could be improved describing the findings obtained also by parents who asked for queries and who refused to have their children immunized - as regards the composition of immunization teams, only nurses are cited as included healthcare workers: Authors could discuss the opportunity to implement these teams with other professions such as physicians
---

VERSION 1 – AUTHOR RESPONSE

Reviewer: 1

Reviewer Name: Suzanne Audrey

Institution and Country: University of Bristol

Please state any competing interests or state 'None declared':

None.

Please leave your comments for the authors below

This paper reports on an important topic, and gives insights into electronic consent for the HPV vaccination programme. The paper is generally well written (I have noted some minor typing errors below) and makes a contribution to knowledge.

However, I do have some concerns which I believe should be addressed before publication.

1. It was not clear to me why the statistical analyses are limited to one cycle, and did not give results for the second year. This is acknowledged as a weakness by the authors and I suspect there is a valid explanation, but it was not clear to me on reading the manuscript.

The statistical analyses were limited to one cycle because the e-consent intervention was not used in the same sample of schools in Year 2 (see response to question 4 for more detail about the schools and related decision-making). Hence, the quantitative part of our evaluation focussed on Year 1 and the qualitative part on Year 1 & 2. The reason for the longer time period for the qualitative part was to capture how the implementers made iterative changes to the intervention over time and as a result of the experience in Year 1.

We have sought to clarify this in the main text in lines 107-109 (description of the intervention) and in the study design section, lines 118-120. I have also added Year 1/year 2 in brackets in different sections of the manuscript to clarify what happened when.

2. The authors indicate e-consent was 'piloted' in June/July 2018 and the quantitative results relate to that year. This almost suggests that we are presented with the results of a pilot study, rather than an evaluation.

Thank you for this observation. We agree that our study focused on evaluating the first iteration of the intervention and have made appropriate changes in the text.

1. Title: Does electronic consent improve the logistics and uptake of HPV vaccination in adolescent girls? a mixed methods theory-informed evaluation of a pilot intervention (page 1, line 4)
2. Abstract: Page 2, line 28, I added pilot. Line 34 states that it was piloted in 14 schools in 2018/19. In line 53 we have also added pilot. (To keep to abstract word count I made edited some words, see tracked changes version.)
3. Strengths and Limitations: Line 62, Page 3 – added pilot
4. Introduction: Line 86 added pilot.

5. Methods: Line 114, added pilot, line 122 added pilot
6. Discussion: Line 395 added pilot.

3. Related to the points above, were data on consent returns and uptake collected for June/July 2019? If so, could these be presented? If not, it would be helpful to be clear why these data were not collected and analysed, as the authors indicate there were a number of problems with the e-consent which may have been less marked as the new system became more familiar and some of the teething problems were addressed.

We did not collect quantitative data on consent form return and uptake in Year 2 (June/July 2019). The quantitative data collected focused only on the schools that participated in the evaluation in Year 1 (June/July 2018).

4. The authors indicate a 'sub-set' of schools took part in June-July 2019. More information on why only a sub-set were involved would be helpful

The Community Health Trust that developed and implemented the intervention decided to use the improved e-consent system in 22 South London secondary schools in June/July 2019. Only one of these schools had taken part in the 2018 pilot, the reasons for using the improved system in this school and not the other pilot schools was due to the scheduling of the HPV vaccination sessions. This was not intentional but as a result of when the improvements to the e-consent intervention post Year 1 experience were ready to implement and how this tallied with the scheduling of booked vaccination sessions. The Trust also wanted to expose additional schools to the e-consent system.

See Lines 191-199, pages 8 & 9 in the methods section

5. The authors indicate they could only measure the difference in the return of consent forms for 22 matched schools. It would be helpful to explain why.

This because there was missing data about consent form returns in three of the intervention schools, which resulted in these 3 schools and their matched pairs being excluded from this analysis. Please see lines 154-155, page 7 and 239, page 11.

6. In relation to vaccine uptake, the authors indicate that those who were absent on the day were vaccinated later bringing the final uptake rate up from around 80% to 86%. Do we know if the 6% used e-consent or paper consent?

Paper consent would have been used in the 14 control schools at school and community based catch up clinics.

All parents in e-consent schools would have received the weblink to the e-consent portal and in most cases will have used this link to complete the consent form. Some students who attended e-consent school catch up clinics did not have a signed e-consent form. In those instances, the nurses would phone parents during clinics or obtain adolescent self-consent if appropriate. In the case of community-based catch up clinics parents/guardians and students would usually attend together, if

the e-consent had not been completed nurses would either complete the e-consent form at the clinic with the parent and student, if Wifi/technology was available or complete a paper consent form.

We have added this as an Endnote due to word count restrictions, this is referenced in line 248, page 11.

7. The authors indicate 'a few schools declined to use the intervention due to concerns about communication with parents. How many schools? Could the reasons be expanded upon as it is an important issue in relation to e-consent.

At the time HRCH did not record the exact number of schools that declined the e-consent intervention. We do know that this type of decline was more common in schools located in the two inner London boroughs. We acknowledge that that lack of numbers of school of schools that declined the intervention is a limitation (see line 70, page 3) and have added more detail about the reasons for decline in the qualitative results section (page 12, lines 279-282, page 12). Concerns about: i) pre-existing barriers to electronic communication with parents, ii) whether a change to consent processes would reduce the return of consent forms and uptake of HPV vaccine, iii) adapting to a new way of working.

8. The authors indicate that, as a result of some of the lessons learned, changes were made to the e-consent intervention, but we do not know if these changes meant e-consent was more effective. This comes back to the issue of whether this study is better described as a pilot study rather than an evaluation.

We agree, and as stated in our response to Question 2 we have made relevant changes.

9. The authors indicate the study addresses an evidence gap. It certainly raises some key issues but I'm not sure if the evidence gap is fully addressed.

We agree, so have changed the wording to 'starts to address'. Line 418, page 18.

10. I spotted typing errors in the abstract and on pp. 6,13, 17. The article would benefit from proof reading to address these minor issues.

The manuscript has been proof-read to address these types of errors.

Reviewer: 2

Reviewer Name: Cecilia Trucchi
Institution and Country: A.Li.Sa., Liguria region, Italy

Please state any competing interests or state 'None declared':

None declared

Please leave your comments for the authors below

In addition to the abovementioned comments, I suggest the Authors to address the following minor issues:

1. The electronic consent intervention was addressed to adolescent girls and to their parents:

what about the gender-neutral HPV immunization strategy in UK?

The universal (gender-neutral) HPV immunisation was introduced in October 2019, so after this evaluation of a pilot e-consent intervention. The findings from this evaluation are likely to be of relevance for the use of electronic consent in other adolescent vaccination programmes including the universal HPV vaccine programme.

2. The matching between healthcare and school's data to verify that all subjects for whom the HPV vaccine is recommended were invited to receive HPV vaccine could be better clarified and deepened.

In England adolescent HPV vaccinations are routinely administered as part of the schoolbased immunisation programme. Health organisations (like HRCH) that are commissioned to deliver this programme work with schools to identify all adolescents, who are eligible for vaccination. These adolescents are then invited to participate in the HPV vaccination programme. Adolescents, who do not attend schools (e.g. home schooling) are followed up separately and mainly encouraged to have the vaccine at the General Practice (primary care health centre) where they are registered. School-aged vaccinations are not routinely offered at General Practices but can be obtained there for free in cases where adolescents have missed the school immunisation and catch-up sessions.

We have not expanded in detail on all these delivery mechanisms in this manuscript since our focus is on the improvement of consent form return and vaccine uptake for vaccines delivered in schools.

In line 78-79, page 4, I have added that immunisation teams and schools identify all students who are eligible for vaccination.

3. The "outputs" section introduces the concept of "maturity and intelligence" assessment of students whose parents were uncontactable: discussing the role of an adolescent "informed assent" in addition/alternatively to the informed consent of parents/tutor could be of interest, since many countries includes these procedures in their immunization process; furthermore, the informed consent's collection of one or both parents/tutors should be better clarified, finally, what about the managing of refusal? Were parents invited to sign an "informed dissent" or similar?

Thank you for this feedback, we are very interested in this concept and have added some additional comments and a text box to the results (see pages, 329-338, pages 13 & 14), which has had some implications for the word count. We are considering writing an additional manuscript to expand on this topic in greater depth. The paper and e-consent forms allowed parents to record whether they agreed or declined vaccination (see line 39, page 2 for e-consent vaccine agreement or decline and line 81-82, page 4 for paper consent agreement or decline).

4. The Results section could be improved describing the findings obtained also by parents who asked for queries and who refused to have their children immunized.

We have added a section on this on page 13, lines 312-317.

5. As regards the composition of immunization teams, only nurses are cited as included healthcare workers: Authors could discuss the opportunity to implement these teams with other professions such as physicians.

In England adolescent immunisations are mainly delivered within schools and school-aged immunisation is led by nurses (see lines 76, page 4). Lessons learned from this evaluation, are however likely to be of relevant to adolescent vaccination programmes led by other health professions, including medical doctors

VERSION 2 – REVIEW

REVIEWER	Suzanne Audrey University of Bristol, UK
REVIEW RETURNED	31-Aug-2020

GENERAL COMMENTS	I believe this manuscript is interesting and makes a contribution to knowledge. I have recommended acceptance but there are two points I would like to make. 1] The way in which the main statistical result is presented was confusing to me. The result states: the significant difference in consent form return rate between e-consent and paper consent schools (73.3% (n=11) vs 91.6% (n=11), p=0.008) response rate (73.3%). I am not a statistician and so this could be my lack of knowledge but it intuitively looks odd to have n=11 repeated after two different percentages. 2] There are still minor typing errors which are distracting. Hopefully, these will be corrected at the proofing stage. For example:  - line 39: agreed to declined - line 196: from a school who a school - line 382: in collection paper - line 416: demonstrates shows
--

VERSION 2 – AUTHOR RESPONSE

On behalf of my co-authors I am very grateful for your review our of manuscript. I have made the suggested revisions and hope that the manuscript is now clearer.

The following changes have been made:

1. Removed the two mentions of n=11 and instead added a sentence that makes it clear that only data from 22 match schools was analysed for this result (see lines 236-239).
2. We have resolved the typing errors that were highlighted by the reviewer. The changes can be found in the following lines: 39, 198, 384 & 418.

I have uploaded new clean and marked versions of the manuscript to the portal and we are very grateful for your consideration of our paper.